# Production of Thermoplastic Starch-*Aloe vera* Gel Film with High Tensile Strength and Improved Water Solubility

**DOI:** 10.3390/polym14194213

**Published:** 2022-10-08

**Authors:** Siti Fatma Abd Karim, Juferi Idris, Junaidah Jai, Mohibah Musa, Ku Halim Ku Hamid

**Affiliations:** 1School of Chemical Engineering, College of Engineering, Universiti Teknologi MARA (UiTM), Shah Alam 40450, Selangor, Malaysia; 2School of Chemical Engineering, College of Engineering, Universiti Teknologi MARA (UiTM), Sarawak Branch, Samarahan Campus, Kota Samarahan 94300, Sarawak, Malaysia

**Keywords:** polymer characterization, thermoplastic starch, *Aloe vera* gel, crosslinker, biodegradable film

## Abstract

Biodegradable film packaging made from thermoplastic starch (TPS) has low mechanical performance and high water solubility, which is incomparable with synthetic films. In this work, *Aloe vera* (AV) gel and plasticized soluble potato starch were utilised to improve the mechanical stability and water solubility of TPS. Dried starch was mixed with glycerol and different AV gel concentrations (0% to 50%). The TPS + 50% AV gel (30 g TPS + 15 g AV gel) showed the best improvement compared to TPS alone. When compared to similar TPS films with AV gel added, this film is stronger and dissolves better in water. Mechanical qualities improved the tensile strength and Young’s modulus of the TPS film, with 1.03 MPa to 9.14 MPa and 51.92 MPa to 769.00 MPa, respectively. This was supported by the improvement of TPS water solubility from 57.44% to 46.6% and also by the increase in decomposition temperature of the TPS. This promises better heat resistance. The crystallinity percentage increase to 24.26% suggested that the formation of hydrogen bonding between TPS and AV gel enhanced crosslinking in the polymeric structure. By adding AV gel, the TPS polymeric structure is improved and can be used as a biodegradable food-packaging film.

## 1. Introduction

Nowadays, plastics are widely used for various applications, depending on their properties and characteristics [1]. Plastics cause disposal problems for the environment due to their non-biodegradable properties. It is estimated that by 2050, nearly 12,000 MT of plastic trash could wind up in landfills or the environment unless significant action is taken in worldwide plastic manufacturing and waste management [2]. Polyethylene, a common synthetic-based polymer, is one of the primary raw materials used to produce plastics. It has poor biodegradability and has become the main environmental threat [3]. Thus, biopolymers were seen as an alternative to producing plastic and have garnered much attention. Starch is the most common biopolymer feedstock for producing films and coatings [4] and is extensively covered in the literature as the primary material for the implementation of active food packaging [5,6,7]. According to Charles et al. [8], potato peel starch can be used in the packaging of chilled and frozen foods. Meanwhile, Thodanakasem et al. [9] used watermelon rind. Morphological structure, mechanical properties, thermal properties, barrier properties, and biodegradability were considered the standard characterization techniques for obtaining a high-quality starch-based film [1,10,11,12,13,14].

Starch degrades faster, thus reducing the need for landfills compared to synthetic-based plastics [15]. However, the stability and the ability of the starch to be processed are limited by adhesion and the creation of merging layers. The instability occurs due to the strong hydrophilicity of starch. Furthermore, due to its fragility, it cannot be utilized for food packing [16]. The combination of starch and a plasticizer produced a thermoplastic starch (TPS), which has enhanced mechanical and barrier properties compared to starch alone. Nevertheless, TPS cannot compete with synthetic polymer-based films. The inferior properties of TPS, such as poor mechanical qualities, low heat stability, and being very sensitive to the presence of moisture, make it unsuitable for food packaging compared to synthetic-based polymers [17]. As a result, several methods have been used to improve TPS, such as changing the way starch additives are made chemically and mixing TPS with different polymers. However, TPS crosslinks between starch and plasticizer result in poor mechanical characteristics and high-water solubility.

According to Bangar et al. [4], crosslinking is a way to strengthen TPS by changing the starch granules’ behaviour. This stabilizes and strengthens the structure of the granules and makes TPS stable at high temperatures. The crosslinker enhances the starch-based film’s properties [18]. Crosslinking agents are used in starch films and most polymeric materials to improve their mechanical properties [19]. This had been proven by Peidayesh et al. [20], where epichlorohydrin was used as a crosslinker in starch and increased the tensile strength. Other types of crosslinkers in starch also used were hexametaphosphate, epichlorohydrin, glutaraldehyde, boric acid, and citric acid [20,21,22,23,24,25]. However, glutaraldehyde and epichlorohydrin are not safe for human consumption due to their inflammatory properties [26]. The addition of citric acid in starch has a minimal effect on strength [27], whereas boric acid makes the material less resistant to water and less tough [28]. Thus, AV gel, which consists of several types of organic acid was suggested in this study to investigate its crosslink formation. Few researchers have applied the embedment of *Aloe vera* into a starch film to enhance its performance, whether as a crosslinker, additive, or modifier [29,30,31,32,33,34].

*Aloe vera* (AV) gel is a biopolymer assumed to contain functional and potential components for use as a coating material that creates a wax-like edible coating [35]. Gutierrez and Gonzalez [33] mentioned that the organic acid and polyphenolic chemicals in AV gel improved the crosslinking between starch molecules, which increased the mechanical characteristics of film made from starch. The AV gel caused changes in mechanical, chemical, and physical properties and improved the antibacterial properties of potato starch/chitosan film [31]. Pinzon et al. [32] discovered that the starch-chitosan interaction was altered by AV gel, with crosslinking occurring between phenolic compounds in AV and starch molecules, affecting the film’s characteristics. The tensile strength of TPS film with AV has a vast gap value. Pinzon et al. [32] found that the tensile strength of plasticized plantain starch with chitosan decreased from 9.7 to 4.6 MPa with 200% to 1000% AV gel in the formulation. Gutierrez and Gonzalez [33] blended plasticized native plantain flour with 2–6% AV gel and found the tensile strength to be 1.43–2.1 MPa, with the Young’s modulus between 82 and 190 MPa. Nieto-Suaza et al. [36] applied AV gel to banana starch at 16.67 times greater than the banana starch content and measured a tensile strength of 3.74 MPa. There is a need to explore different AV gel concentrations that impact the mechanical properties of films. The area of mechanical properties also needs further exploration, since a vast gap was observed in the concentration of AV gel applied to the plasticized starch-based film. In addition, most studies used the film-forming solution method in the production of the film, which does not represent actual industry practices.

The same knowledge gap is present in the water solubility of starch-based films with AV gel. Nieto-Suaza et al. [36] recorded 11.8% water solubility for 50% AV gel in 3% starch, Pinzon et al. [32] obtained 36.3–45.2% water solubility in 200–1000% of AV gel concentration, while Gutiérrez and Álvarez [34] discovered 55–57% water solubility for 2–6% AV gel in starch/chitosan-based film. Further exploration of the water solubility of the film is needed because of the difference in percentages among the abovementioned studies. In addition, none of them covered the water absorption capacity of the prepared film. Plus, most studies used the film-forming solution method in the production of the film, which does not represent actual industry practices. A very limited number of studies reported on investigating the impact of AV gel on the characteristics of starch-based film. The presence of hydrogen bond acceptor and donor in the chemical structure of AV gel is expected to give a strong crosslinking formation between AV gel and TPS polymeric structure, possibly improving the mechanical properties and water solubility of TPS film.

Therefore, to the authors’ knowledge there is still much work left to explore the impact of AV gel incorporation on the mechanical properties and water solubility of TPS-based films. Previous studies recorded a vast difference in the mechanical properties, and a limited report discusses water solubility performance in a film incorporated with AV gel. Thus, the AV ratio in TPS-based film has become an essential subject matter to further explore. The key idea in this paper was to use 10 to 50% AV gel in TPS film to explore AV impacts on mechanical properties and water solubility performance. The samples were prepared by using melt-blending and hot-pressing techniques. The AV gel functioned as a crosslinker and was then investigated on other characteristics: thermal decomposition, thermal properties, crystallinity, colour, thickness, morphological structure, and functional compound.

## 2. Materials and Methods

### 2.1. Thermoplastic Starch Preparation

Powdered soluble potato starch (Bendosen) was obtained from Laupik Chemical. The starch was dried in a universal oven (ULE 600, Memmert) at 70 °C for 24 h to remove moisture until the moisture content was 5%. Then, 21 g starch was mixed with 9 g glycerol (MW: 92.09; Chemiz (M) Sdn. Bhd.) at a 70:30 ratio to form TPS. The TPS was then sealed and stored in a desiccator for 24 h.

### 2.2. Film Preparation

The film was prepared based on a modified melt-blend and hot-press technique previously published [37]. AV gel was obtained from Chemieconnex, Malaysia. The gel was added to TPS at different concentrations (3, 6, 9, 12, and 15 g) into an internal mixer (Haake Polylab OS RheoDrive7, Thermo Scientific) at 170 °C, 60 rpm, and 30 min to produce solid resin. The solid resin was formed once the mixture was removed from the internal mixer at room temperature. Afterward, the solid resin was crushed using a compact crusher (HMRV50-19, Rexmac) to form a smaller resin size. The crushed resin was then subjected to a hot-pressing procedure using a hot press machine (QC-602A, Cometech) to convert the resin into a thin film. Six different samples were prepared: control, TPS, and a combination of TPS with different AV gel concentrations (10%, 20%, 30%, 40%, and 50% over the total weight of TPS).

### 2.3. Characterization of the Biodegradable Film

#### 2.3.1. Mechanical Properties

The solid film samples of TPS and TPS with AV gel that had been made were cut into pieces that were 100 mm long and 25 mm wide, which was the size recommended by ASTM D882. A universal testing machine (H50KT, Instron 3382, Tinius Olsen, Surrey United Kingdom) was used to measure the mechanical properties of the samples (e.g., tensile strength, elongation at break, and Young’s modulus). The strip specimens were strained at a 25 mm/min rate at room temperature and relative humidity of 50 ± 5% [38].

#### 2.3.2. Water Absorption and Water Solubility

For water absorption and solubility analysis, the film samples were cut in a square dimension of 2 × 2 cm and dried in a universal oven (ULE 600, Memmert) for 24 h at 50 °C. The sample was weighed (*W*_0_). The samples were then immersed in a beaker containing 250 mL of distilled water for 24 h, drained, and weighed again (*W*_1_). Afterward, the sample underwent a second drying process in an oven at 50 °C for 24 h, and the sample was weighed (*W*_2_). The percentage of water absorption was measured using Equation (1), and water solubility was measured using Equation (2). The method was based on ASTM D570-63 [39].
(1)$$Water\;absorption\;\left(\%\right)=\frac{W_{1}-W_{2}}{W_{0}}\times 100$$
(2)$$Water\;solubility\;\left(\%\right)=\frac{W_{0}-W_{2}}{W_{0}}\times 100$$

#### 2.3.3. Thermal Decomposition

Thermal decomposition properties were obtained from a thermal gravimetric analyser (TGA; SDTA581e, Mettler Toledo, Columbus, OH, USA) under a 50 mL/min nitrogen atmosphere with samples of approximately 20 mg, with a heating rate of 10 °C/min for a temperature range from 25 to 600 °C [40].

#### 2.3.4. Thermal Properties

Differential scanning calorimetry (DSC; Star^e^ system, Mettler Toledo) was then used to measure the melting temperature (T_m_) of the sample. The samples were cut and prepared in encapsulated aluminium pans at 5.2–5.5 mg weight. The scans were performed in one single process, heated from 25 to 350 °C. The setup used was a nitrogen atmosphere flow rate of 50 mL/min with a heating rate of 20 °C/min.

#### 2.3.5. Crystallinity

The crystallinity was identified using X-ray diffraction (XRD; D/Max 2200V/PC, Rigaku, Tokyo, Japan). The range of diffraction angles (2θ) was 5°–40°, with a scan speed of 5°/min, 40 kV, and 30 mA, which were similar to the operating conditions performed by Ramírez-Hernández et al. [41]. The crystal structure of the material was determined, and the percentage of crystallinity was calculated based on Equation (3).
(3)$$Crystallinity\;\left(\%\right)=\frac{Area\;of\;crystalline\;peaks}{Area\;of\;total\;peaks\;\left(crystalline+amorphous\right)}\times 100$$

#### 2.3.6. Visual Appearance and Thickness

The visual appearance of the films was photographed using a commercial smartphone camera (Redmi Note 9 Pro, Xiaomi, Beijing, China) [42]. The thickness was measured using a digital thickness gauge (Mitutoyo 700-118-30), ranging from 0 to 12 mm. The reading came nearest around 0.1 μm at eight random locations, and the average thickness value was obtained. All films were preconditioned by placing them in a desiccator for at least 24 h before further testing.

#### 2.3.7. Morphological Structure

Scanning electron microscopy (SEM; JSM-7600F, JOEL) was used to evaluate the surface morphological structure of the film at 500× magnification. Using a two-sided adhesive carbon model, a 4 × 4 mm sample film was directly placed on a 12.5 mm diameter aluminium holder. The sample was then coated with platinum using a sputter coater (Polaron SC 7620). A low vacuum was used to introduce the prepared sample into the microscopy chamber.

#### 2.3.8. Functional Compound

The functional groups were observed via Fourier transform infrared spectroscopy (FTIR) spectrometer (Spectrum One, PerkinElmer, Waltham, MA, USA). The spectrum range was between 400 and 4000 cm^−1^ with 4 cm^−1^ resolution and 64 scanning times.

#### 2.3.9. Statistical Analysis

Statistical analysis was performed with Microsoft^®^ Excel^®^ for Microsoft 365 MSO. A one-way analysis of variance (ANOVA) followed by post hoc *t*-test were carried out to detect significant differences between the films’ properties. The significance level used was *p* < 0.05, but with Bonferroni adjustment was *p* < 0.0083.

## 3. Results

### 3.1. TPS Process Condition

Torque performance is a TPS process condition that reveals crucial TPS characteristics for ensuring mixture homogeneity. Figure 1 shows the constant torque values of TPS film and TPS + 10–50% AV gel versus time, indicating that each sample was mixed uniformly within 30 min. Considering this, the mechanical, chemical, and physical properties of TPS-based film with various concentrations of AV were investigated in greater detail.

### 3.2. Mechanical Properties

Table 1 shows that the tensile strength of the TPS film increased with the higher content of AV gel. The highest tensile strength was recorded at TPS + 50% AV with 9.14 MPa, an increase of 791% from the tensile strength of TPS film without additional AV gel (control). Polysaccharides and phenolic compounds in AV gel led to covalent and non-covalent bonding between starch and glycerol. The phenolic compound from the citric acid in AV gel worked as a crosslinker, creating a covalent bond between the starch and AV gel. Thus, the intermolecular bond becomes stronger, and films become more resistant to the stress given, increasing the tensile strength of the TPS. This result is consistent with Kanatt and Makwana [43], who suggested that citric acid, which consists of hydroxyl and carbonyl groups, acted as a crosslinker in carboxymethyl cellulose film, which led to stronger covalent bonds between the molecules.

Meanwhile, the addition of AV gel in TPS film caused a declining trend in elongation at break, which is listed in Table 1, column 3. The highest elongation at break of TPS film was at 11.26%, but it decreased to 5.63% due to the addition of 10% AV gel. This is because the crosslinker reinforcement inhibits chain mobility, thus decreasing the elongation at TPS break. This could be related to the increasing internal hydrogen bonds between polymer chains and the corresponding decrease in molecular volume, which enhances brittleness and decreases elongation at break [44]. This result agrees with Pinzon et al. [32], who found a similar trend where elongation at break decreased as the concentration of AV gel in the banana-starch edible films increased. However, the elongation at break remained almost constant (1.62% to 2.11%) with more than 20% AV gel, which indicates no significant changes had occurred.

As tabulated in Table 1, column 4, Young’s modulus shows similar trends with the tensile strength. The considerable increase in Young’s modulus following the addition of AV gel to TPS film ranged from 51.92 to 769.00 MPa for TPS film containing 50% AV gel. This occurred due to the formation of crosslinking between AV gel and TPS. Therefore, it is suggested that the AV gel impedes the motion of dislocations through the lattice in TPS, which increases the Young’s modulus value. This result is in line with Paiva et al. [45], who found that the interaction of nanofibers with a plasticizer enhanced the effect of the plasticizer and increased the Young’s modulus of the material.

### 3.3. Water Absorption versus Water Solubility Percentage

Figure 2 shows the water absorption and water solubility percentage of TPS film with different AV gel percentages. The water absorption percentage increased from 134.33% to 326.46% with the increasing concentration of AV gel. The films are attracted to water molecules due to the hydrophilic properties of AV gel, which enhances the surface affinity of the film. The hydrophilic character of AV improved the capacity of the film to absorb and retain water and simultaneously increased the weight of the TPS. In addition, the amount of end hydroxyl groups and the hydrophilic nature of AV indicate water sensitivity, as explained by Bulatović et al. [46].

For the water solubility, the percentage decreased from 57.44% to 46.65% with the addition of AV gel in TPS film. This result shows significant water resistance improvement because the solubility decreased by 10.79% upon adding 50% AV gel to TPS. The reduction of water solubility might be attributed to AV gel components such as polysaccharides and phenolic compounds in the entrapped thermoplastic starch matrix, which create a strong network of starch–glycerol interaction. The hydrophobic area in the polyphenolic area in the AV gel causes strong starch–glycerol interaction, thus reducing the solubility percentage. The crosslinking formation in the polymer matrix inhibits the film from having greater dispersion caused by water penetration at higher AV gel concentrations. The interfaces enhance the cohesive properties of the polymer matrix and prevent water molecules from breaking these strong bonds. This argument is consistent with a previous study’s finding that AV increased the polarity of starch/chitosan blends and reduced water solubility [31]. This result is also corroborated by a finding where the presence of AV gel reduced the water solubility of plain chitosan film from 100% to 45.12% [47].

### 3.4. Thermal Degradation

Figure 3a depicts the thermogravimetric analysis of TPS film with varying AV gel concentrations, whereas Figure 3b depicts the derivative thermogravimetric (DTG) analysis of the films. Thermogravimetric analysis (Figure 3a) shows that the initial weight reduction of TPS film from 30 to 267.63 °C represents the decomposition temperature of the moisture and glycerol trapped in TPS film. The addition of AV gel led the temperature range to expand from 30 °C to between 281.02 and 286.27 °C, indicating a stronger hydrogen connection between glycerol and starch as a result of crosslinking. The findings were compatible with Boonsuk et al. [48], where the TPS moisture evaporation was between 50 and 130 °C. Meanwhile, Hafila et al. [49] and Nieto-Suaza et al. [36] also supported the result by mentioning that the glycerol-rich phase containing starch decomposed between 125 and 290 °C.

The significant decomposition steps for TPS and TPS + 10–50% AV film were from 267.63 to 322.98 °C and 281.02 to 326.82 °C, respectively. The decomposition temperature range refers to the partial decomposition of starch associated with the breakdown of hemicellulose, amylose, and amylopectin glycosidic linkages. This result agrees with the previous finding stating that starch decomposed at a temperature range between 320 and 330 °C [31,48,49]. The increase in the temperature range represents the crosslinking reactions between starch and AV gel. This temperature range showed the apparent presence of the DTG peak, as shown in Figure 3b. The DTG peaks of all TPS + 10–50% AV gel were higher (310.11–312.29 °C) than those of TPS film (304.67 °C). This indicates that chain restructuring or the formation of new bonds occurred between the TPS and AV gel as a result of the temperature change [31]. These results agree with Gutiérrez and Álvarez [34], who observed a slight increase in decomposition temperature between 320 and 340 °C attributable to a larger glycerol–flour interaction with a greater concentration of AV gel, which represents the decomposition of partially decomposed starch.

Finally, a slow and stable decomposition rate was detected between 330 and 500 °C. The weight loss of TPS + 50% AV film was 44.66%, which was low compared to the TPS residual percentage, which was 53.10%. This is likely due to a decreased proportion of minerals and aromatic ring oxidation products (coal residues) in the AV gel, as also explained by Gutiérrez and Álvarez [34]. TPS with AV has a 46.26% higher residual weight than TPS (30.28%). AV gel influenced the thermal resilience of starch-based samples and significantly slowed the breakdown process compared to pure starch film systems that had not been treated, suggesting better bonding on the TPS film.

### 3.5. Thermal Properties

Figure 4 shows the DSC of TPS film with different AV gel concentrations in all samples. The glass transition temperature (T_g_) for TPS and other films was not seen because most TPSs have a T_g_ of approximately −75 to 10 °C [50]. The addition of AV gel also caused the melting temperature (T_m_) of TPS at 105.85 and 239.99 °C to disappear due to a poor peak. However, the T_m_ of TPS at 302.09 °C was shifted to a higher value of 320.93 °C upon adding 50% AV gel. Peak 320.93 °C represents the melting of crystallized amylopectin and co-crystallized amylose. Crosslinking between starch and AV gel raised the T_m_ and led to extensive interaction between the AV gel and the non-crystalline portion of the starch chain. Thus, the TPS crystallized. This result is in agreement with Amin et al. [51], who found the melting of crystallized amylopectin and co-crystallized amylose to be between 250 and 330 °C.

### 3.6. Crystallinity

The X-ray diffraction of developed TPS film showed different patterns with different AV gel concentrations, as shown in Figure 5. The apparent peak of TPS was observed at 13.6° and 16.8°, representing type V_a_ anhydrous crystal structure, while 19.70° and 20.98° represented type V_h_, hydrated crystal structure [52]. Other researchers have obtained a similar XRD peak for TPS film [30,36,53]. The prominent reflection peaks for TPS crystallinity were 12.8° and 19.6°. Upon adding AV gel, 2θ peaks 13.6° and 20.3° shifted to the left and became 12.2° and 18.62°, proving V-type crystalline structure enhancement in TPS. In addition, these two peaks also experienced intensity increase, suggesting that a stronger hydrogen bonding between glycerol and starch had occurred. This result is also supported by Vedove et al. [54], who obtained a peak of 13.1° for TPS, and its intensity had a small increase with the addition of anthocyanin. The peak of 19° corresponds to the crystalline V-type structure, demonstrating interactions between amylose and glycerol [55].

The intensity reduction of peak 16.8° at 10% AV and its disappearance with higher AV gel concentration in TPS film proved to broaden the XRD pattern affected by the presence of many lattice defects and small crystal size, suggesting changes in structural order. Peak broadening indicates that the interaction between AV and starch–glycerol is successful. Thus, we conclude that the amylose substitution degree increased with a larger AV gel concentration, resulting in a stronger interaction between the glycerol and the amylopectin chains. The crosslinking of amylose chains restricts glycerol–amylose interactions, resulting in amylopectin–glycerol interactions [34]. It is suggested that the crystalline structure forms upon the addition of AV gel, where polymeric carbohydrate chains are produced.

The degree of crystallinity of each film was calculated and recorded as X_c_ in Figure 5. It is indicated as the long-range order in a material that affects the film properties. The X_c_ increased from 7.61% for TPS to 24.46% for TPS + 50% AV gel. This result agrees with previous studies that obtained X_c_ for TPS from about 10% to 14.23% [54,56]. During plasticization, the intermolecular starch and hydrogen bonds between the plasticizer and starch molecules had an effect on the X_c_. Various interactions (e.g., hydrogen bonding) of amylose–amylopectin, amylopectin–amylopectin, and its reactivity with AV led to the creation of such starch ordering patterns during film production. Dagmara et al. [31] stated that AV gel increases starch crosslinking, which results in the alignment of carbohydrate polysaccharide chains and the transformation of the amorphous state into a crystalline one, as demonstrated by X-ray diffraction.

### 3.7. Visual Appearance

Figure 6 displays the physical appearance of TPS and TPS + 10–50% AV gel films. TPS appeared colourless, and upon adding more concentrations of AV gel, the film became increasingly yellow. Adding 40% and 50% AV gel to TPS created a dark brown colour compared to only 10–30% AV gel. The presence of yellowness suggests the presence of crosslinking between the TPS and AV gel. Due to oxidation, the crosslinking degree affects the film colour, resulting in a brownish film. The observations also agree with Pinzon et al. [32], who observed a decrement in transparency due to the increase in solid film content from AV gel. The presence of amino acids in AV gel is suggested to influence the yellowish colour of TPS-based film. As reviewed by Garavand et al. [57], crosslinking of proteins by citric acid results in yellow-brown films.

### 3.8. Thickness

As listed in Table 1, column 5 for the thickness of the TPS film with different AV gel concentrations, the thickness of TPS film increased with AV gel increases. The thickness increased by 129% with 50% AV gel added to TPS film compared to TPS film without AV gel. The presence of AV gel strengthens the internal bonds in starch, increasing the molar volume and influencing the formation of hydrogen bonds between starch and glycerol during gelatinization. Consequently, starch and glycerol formed a stronger interaction that led to the formation of a thicker film during gelatinization. These interactions appeared to grow more robust as the concentration of AV gel rose. This result is also in agreement with the studies by Gutiérrez and Álvarez [34], Kanatt and Makwana [43], and Pinzon et al. [32], where the polyphenolic content in AV gel provided a crosslinking effect that increased the dry matter and the thickness of starch–chitosan interaction.

### 3.9. Surface Morphology

Figure 7 shows the surface morphology of TPS and TPS + 10–50% AV gel that exhibited compact morphology without pores and cracks. TPS and TPS + 10% AV gel showed inhomogeneous morphology with numerous starch grains in the plasticized matrix. The effect was due to the number of grains in the hanging long chains of soluble potato starch, which caused difficulty in plasticizing and forming a homogeneous TPS film. Similar findings were observed on acetylated TPS with low acetylated starches processed by solution casting [58]. TPS + 20% and 30% AV gel showed smoother and more uniform surfaces, indicating that the components dispersed well with each other. However, TPS + 40% and 50% AV gel films showed irregular and rough surface structures. This occurred due to the high matrix chain interaction caused by the inclusion of AV gel. Adding AV gel caused the surface to wrinkle but did not indicate that the film was irregular and disconnected. The wrinkles may be created by film-to-plate adhesion during peeling. These findings are consistent with the study of TPS film with gelatine, in which wrinkles occurred due to adhesion between the film and the plastic plate when peeled off [59].

### 3.10. Functional Compound

Figure 8a shows the FTIR spectra of TPS film, AV gel, and TPS + 30% AV gel film, while Figure 8b displays the FTIR spectra of TPS + 10–50% AV gel films. Figure 8a is divided into four regions. In region I, the addition of AV caused the O–H peak for TPS at 3288 cm^−1^ to become wider and sharper, suggesting that AV water content was entrapped in the film. At the same time, the AV gel peak at 3275 cm^−1^ became smaller and shifted to 3286 cm^−1^ due to water evaporation during the mixing process. The wide wavelength range between 3000 and 3600 cm^−1^ implies a dipole–dipole hydrogen bond formation. This result is supported by a previous study, which mentioned linkages may occur between the same (starch–starch) or different (AV polysaccharide–starch–glycerol) molecules in the examined systems [31]. For region II, the amino acids in AV gel caused peaks of 2920 and 2871 cm^−1^ to shift to 2915 and 2849 cm^−1^, respectively, and to become sharper. The interaction of amino acids with amylose is affected by the length of the amino acid chains in AV gel. Wang et al. [60] found that the same peak shift occurred in potato starch when a fatty acid was added. In region III, the 1647 cm^−1^ peak of AV sharpened, and the 1640 cm^−1^ peak from TPS dampened and shifted to 1646 cm^−1^ after both materials were combined, suggesting a crosslink involving the C=O structure due to the amino acids in AV gel, which increased the peak height. The intense absorption peak observed between the 1690–1750 cm^−1^ region represents the characteristic of the C=O group in citric acid [43]. From the observation in region IV, the combination of TPS and AV caused obvious stronger peaks at 989 and 1000 cm^−1^. Dagmara et al. [31] stated that the intense absorption bands at 900–1200 cm^−1^ are present due to the C–O–C stretching vibration of polysaccharides in AV gel and starch. However, based on Figure 8b, the different percentages of AV gel did not have a significant impact on the FTIR spectrum. Therefore, the addition of AV to TPS shifted some wavenumbers, suggesting some interactions between components. The FTIR analysis shows the crosslinking reaction between the hydroxyl groups of starch, glycerol as a plasticizer, and amino groups in the AV gel in the blend components.

The formation of crosslinking is also shown in Figure 8a by the transmittance percentage of the samples. A higher percentage of transmittance indicates a more light-friendly surface. Similarly, a low percentage transmittance score indicates that the surface is highly reflective and absorbs little light. In each region, adding AV gel to TPS film reduced the percentage of transmittance, proving inhibiting lighter from passing through the samples occurs. In region I, the transmittance peak of 3315 cm^−1^ experienced transmittance of 74.02% reduced to 53.08%; in region II, 87.03% became 73.25%; in region III, 94.63% went down to 90.90%; and region IV, 51.45% underwent significant reduction to 13.08%. Thus, it is suggested that AV gel addition formed a crosslinked structure with TPS to prevent more light from passing through the film at this peak. Dang and Yokyan [61] stated that the O-H bands became broader with lower absorbance intensity, reflecting the reduction of the average strength of the hydrogen bonds and a broadening of the distribution. The absorbance and transmittance are inversely related.

### 3.11. Summary of Results and Data Comparison with Other Studies

Table 2 summarises the analyses performed in this study. The addition of AV to a TPS-based film improved its mechanical properties and water solubility. The addition of 50% AV gel to TPS resulted in the best tensile strength and Young’s modulus and the lowest water solubility percentage, demonstrating crosslinking in the polymer matrix. Crosslinking is also endorsed by the increasing values of thermal decomposition and melting temperature of the polymer matrix. Furthermore, a more substantial crystalline structure, as evidenced by a degree of crystallinity, suggests that the crystalline structural order was improved upon by adding AV into TPS-based film, which supports the rise in tensile strength and Young’s modulus. Furthermore, increased film thickness and yellowness of the film colour significantly support the crosslinking phenomenon with homogenous morphological surface structure.

Table 3 compares the results produced by this study with those obtained by six other almost-identical published papers. The tensile strength achieved is comparable to that obtained by Pinzon et al. [32] but with an astonishingly high Young’s modulus compared to that obtained by Gutiérrez and González [33]. The water solubility is likewise lower than that obtained by Gutiérrez and Álvarez [34], showing good properties, and is comparable to that obtained by Pinzon et al. [32].

## 4. Conclusions and Recommendations

TPS films consisting of starch, glycerol, and AV gel were successfully formulated and fabricated that showed significant improvement in mechanical properties and water solubility. The 30 g of TPS with the addition of 15 g AV gel improved the tensile strength from 1.03 MPa to 9.14 MPa and the Young’s modulus of TPS film from 51.92 MPa to 769.00 MPa. The water solubility of the TPS film also showed improvement, from 57.44% to 46.6%. This occurred due to crosslinking formation suggested by the hydrogen bonding between the OH group in TPS and the double bond O structure in AV gel, which formed a crosslink within the TPS polymeric structure. Thus, it contributed to the formation of the crystalline structure in TPS film, providing higher heat resistance, turning the TPS colour to brown, and increasing the film’s thickness. In this work, the crosslinking improved the moisture barrier and has comparable mechanical qualities to be proposed for food simulation testing, suggesting the product has potential food packaging application.

## Figures and Tables

**Figure 1 polymers-14-04213-f001:**
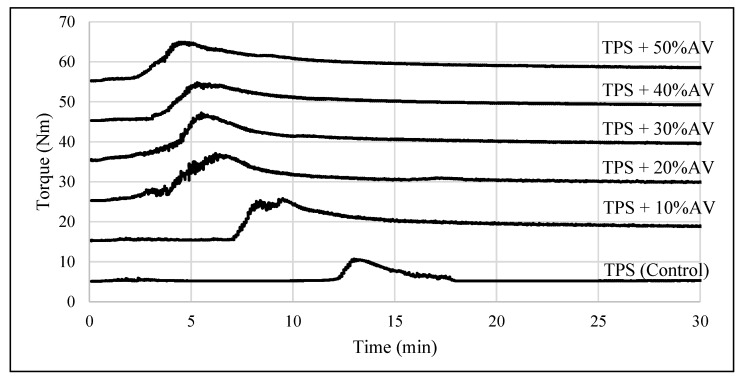
The torque value of TPS film with different AV gel concentrations.

**Figure 2 polymers-14-04213-f002:**
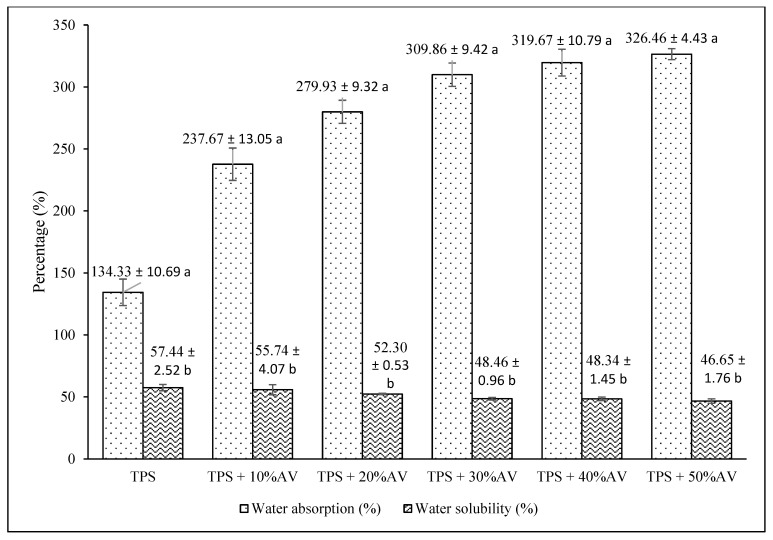
The percentages of water absorption and water solubility of TPS film with different AV gel concentrations. Different letters indicate significant differences between the values at the level of significance, *p* < 0.0083 (Bonferroni adjustment), one-way ANOVA, post-hoc *t*-test. Values are mean ± SD (standard deviation).

**Figure 3 polymers-14-04213-f003:**
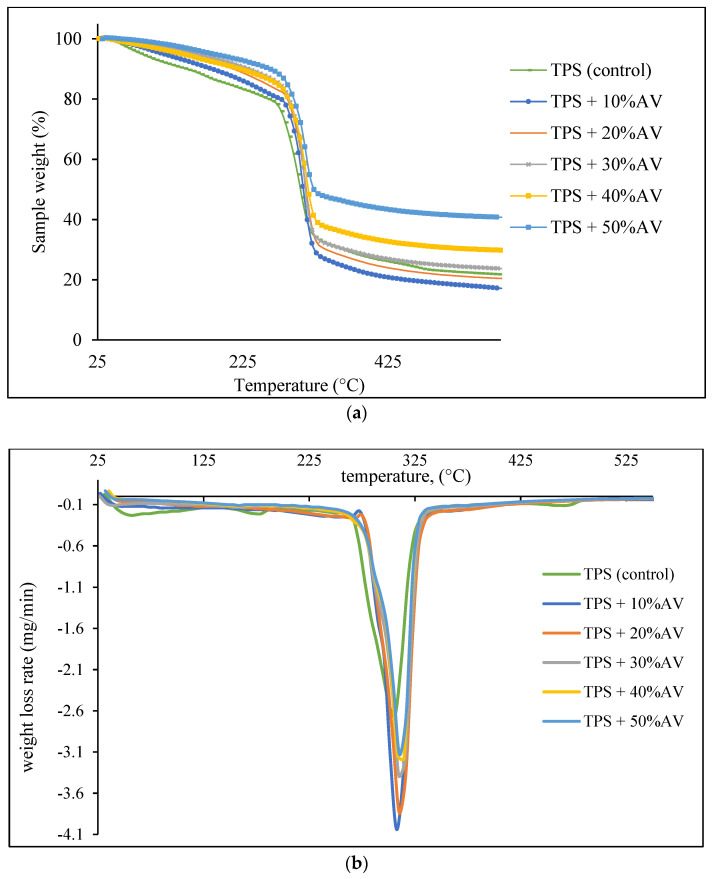
(**a**) The thermogravimetric and (**b**) the derivative thermogravimetric (DTG) analyses of TPS film with different AV gel concentrations.

**Figure 4 polymers-14-04213-f004:**
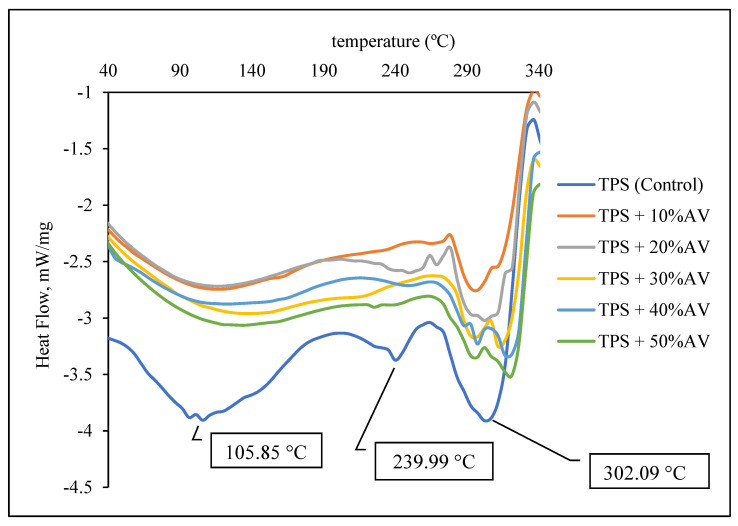
DSC of TPS film with different AV gel concentrations.

**Figure 5 polymers-14-04213-f005:**
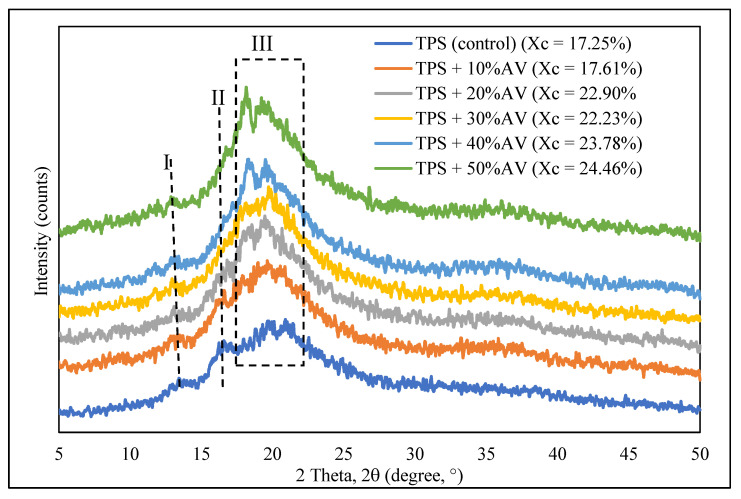
X-ray diffraction pattern of TPS film with different AV gel concentrations.

**Figure 6 polymers-14-04213-f006:**
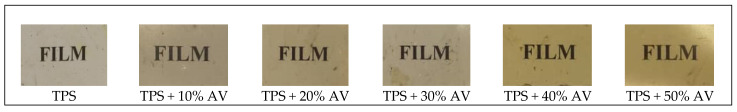
The physical appearance of TPS film with different AV gel concentrations.

**Figure 7 polymers-14-04213-f007:**
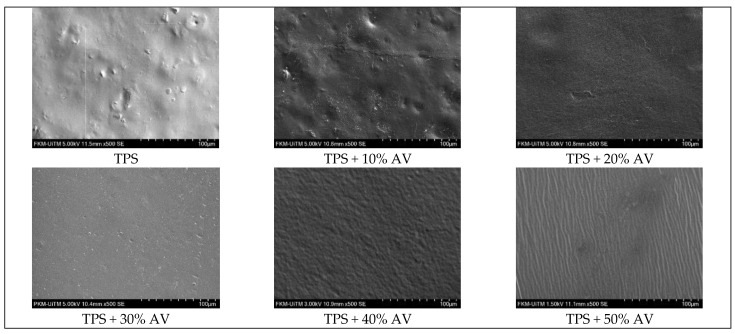
The morphological structure of TPS film with different AV gel concentrations at 500× magnifications.

**Figure 8 polymers-14-04213-f008:**
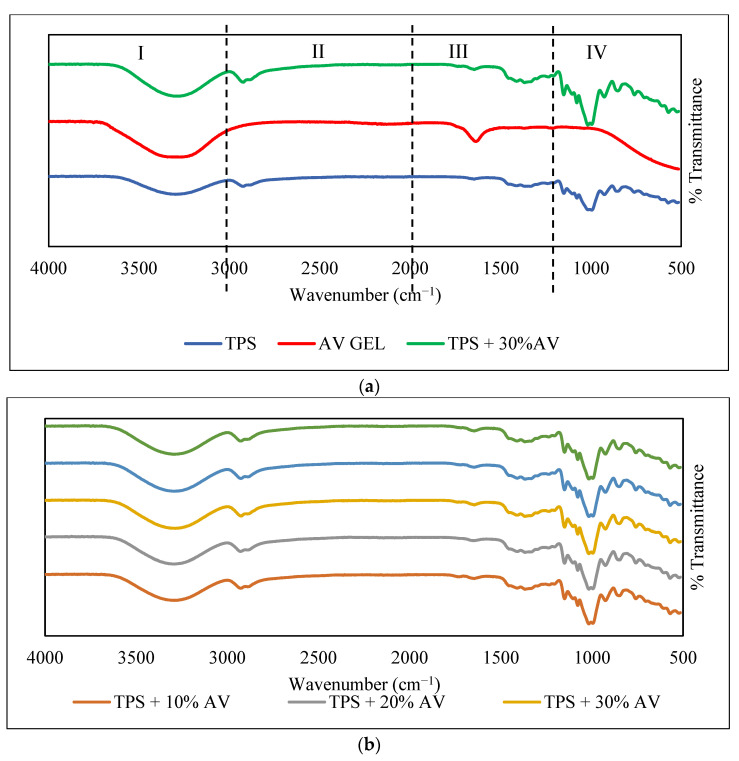
(**a**) The FTIR spectra for TPS, AV gel, and TPS + 30% AV and (**b**) the FTIR spectra of TPS film with different AV gel concentrations.

**Table 1 polymers-14-04213-t001:** The tensile strength, elongation at break, Young’s modulus, and thickness of TPS film at different concentrations of AV gel.

SAMPLE	TS ± SD	EAB ± SD	YM ± SD	Thick ± SD
TPS (control)	1.03 ± 0.34 ^a^	11.26 ± 1.88 ^a^	51.92 ± 15.50 ^a^	0.2342 ± 0.03 ^a^
TPS + 10% AV	4.48 ± 1.68 ^b^	5.63 ± 1.87 ^b^	189.79 ± 106.49 ^b^	0.4357 ± 0.08 ^b^
TPS + 20% AV	6.71 ± 1.90 ^c^	1.62 ± 0.48 ^c^	443.04 ± 284.27 ^c^	0.4471 ± 0.08 ^c^
TPS + 30% AV	7.78 ± 1.67 ^d^	2.11 ± 0.99 ^d^	504.33 ± 174.98 ^d^	0.4484 ± 0.06 ^d^
TPS + 40% AV	8.96 ± 1.80 ^e^	1.83 ± 0.82 ^e^	687.67 ± 159.46 ^e^	0.5094 ± 0.07 ^e^
TPS + 50% AV	9.14 ± 1.47 ^f^	2.02 ± 0.99 ^f^	769.00 ± 88.07 ^f^	0.5127 ± 0.05 ^f^

TS—mean of tensile strength ± SD, EAB—mean of elongation at break ± SD, YM—mean of Young’s modulus ± SD, WA—mean of water absorption ± SD, WS—mean of water solubility ± SD, thick—mean of thickness ± SD. Different letters indicate significant differences between the values at the level of significance, *p* < 0.0083 (Bonferroni adjustment), one-way ANOVA, post-hoc *t*-test. Values are mean ± SD (standard deviation).

**Table 2 polymers-14-04213-t002:** Summary of results.

Characterization	TPS Film	TPS/10% AV–TPS/50% AV
Tensile strength, MPa	1.03 ± 0.34 ^a^	4.48 ± 1.68 ^b^–9.14 ± 1.47 ^f^
Elongation at break, %	11.26 ± 1.88 ^a^	5.63 ± 1.87 ^b^–2.02 ± 0.99 ^f^
Young’s modulus, MPa	51.92 ± 15.50 ^a^	189.79 ± 106.49 ^b^–769.00 ± 88.07 ^f^
Water absorption, %	134.33 ± 10.69 ^a^	237.67 ± 13.05 ^b^–326.46 ± 4.43 ^f^
Water solubility, %	57.44 ± 2.52 ^a^	55.74 ± 4.07 ^b^–46.65 ± 1.76 ^f^
Maximum temperature, T_max_, °C	304.87	307.75–312.29
Melting temperature, T_m_, °C	306.89	307.01–320.93
Crystallinity, %	17.2458	17.6077–24.4592
Thickness, mm	0.2342 ± 0.03 ^a^	0.4357 ± 0.08 ^b^–0.5127 ± 0.05 ^f^
Physical appearance	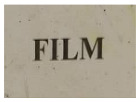	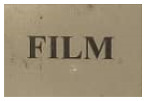 – 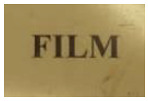
Morphology	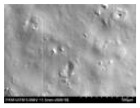	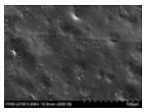 – 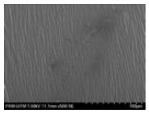
Functional group	3288 cm^−1^ (broader and sharper) 3275 cm^−1^ (slighter) 2920 and 2871 cm^−1^ (shifted) 1647 cm^−1^ (sharpened) 1640 cm^−1^ (dampened) 1646 cm^−1^ (shifted) 1410 cm^−1^, 1366 cm^−1^, and 1244 cm^−1^ (stronger)	

Different letters indicate significant differences between the values at the level of significance, *p* < 0.0083 (Bonferroni adjustment), one-way ANOVA, post-hoc *t*-test. Values are mean ± SD (standard deviation).

**Table 3 polymers-14-04213-t003:** Comparison with other studies.

Sources	Method	Formulation	TS (MPa)	EAB (%)	YM (MPa)	TGA (°C)	T_m_ (°C)	Crystallinity (%)	Thickness (mm)	Water Absorption (%)	Water Solubility (%)
This paper	Melt-blend and hot-press technique	Potato starch + glycerol + AV gel: 10–50%	9.14	2.11	769	304.87 increased to 307.75–312.29	Peak shift from: 105 to 134, 306 to 320. Peak disappeared: 239.99	24.46	0.52	326.46	46.65
[34]	Film-forming solution	Plantain flour + glycerol + AV gel: 0–6%	-	-	-	100, 160–290, 330	-	9–16	0.017–0.042	-	55–57
[33]	Film-forming solution	Plantain flour + glycerol + AV gel: 0–6%	1.43–2.1	24–31	82–190	-	-	-	-	-	-
[30]	Casting	Corn starch + glycerol + AV: starch: 1:3, 1:2, and 1:1	-	-	-	-	-	6.8–13.1	0.064–0.067	-	-
[32]	Film-forming solution	Plantain starch + chitosan + sorbitol + AV gel: 200–1000%	9.7–4.6	30.3–10.4	-	-	-	-	0.0735–0.1745	-	36.3–45.2
[36]	Casting	Banana starch + glycerol + AV gel + different types of curcumin	3.74–5.01	45.4–56.3	-	70, 201, 300	-	No significant effect	0.096–0.1404	-	11.8–32.9
[31]	Film-forming solution	Potato starch + chitosan + AV gel: 10–50%	-	-	-	First stage: 78–95 Second stage: 247, 278, 303	-	25.63–33.71	-	-	-

## Data Availability

The raw/processed data required to reproduce these findings cannot be shared at this time as the data also form part of an ongoing PhD study.

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
