# Peer review of "Production of Thermoplastic Starch-Aloe vera Gel Film with High Tensile Strength and Improved Water Solubility"

_polymers, 2022, doi:10.3390/polym14194213_

Round 1
Reviewer 1 Report
Detailed recommendation:
Key words: please add: biodegradable film
In which year you made this experiment?
In how many repetition you made yours films?
What statistical methods were used?
Figure 8. Check units
Paragraph 3.10. Check references
Did the results differ statistically significantly?
Author Response
Dear Editor,
The coauthors and I greatly appreciated the reviewer's encouraging, critical, and constructive comments on this manuscript. The comments were highly detailed and beneficial in enhancing the manuscript. The manuscript has been revised as per the comments given by the reviewer.
Please see the attachment

Reviewer 2 Report
Comments to Authors
Article “Production of thermoplastic starch-Aloe vera gel film with high tensile strength and improved water solubility” highlights the applications of aloe-vera gel in starch films formation. I think that authors should try to include a scientific discussion, include a clear comparison between samples (i.e., at least indicating the corresponding advantages), and write clearer conclusions. Different points must be considered before the paper can be considered for acceptance.
1. The Introduction needs to be improved. The background about starch films is insufficient and more references should be included in other suitable papers about starch film applications and characterization.
2. Add the justification in Introduction section why this study was conducted.
3. In section 2.2., justify why solid resin was used rather than direct gel.
4. Section 2.3 Characterization of Biodegradable Film; separate the methods with separate headings.
5. To describe the range use lesser no first and larger number later (11-100% rather than 100-10%).
6. Maximum data is shown in figures, convert some data into Tables.
7. Also add the data for transmittance, which is very useful for deciding the role of films.
8. Conclusion is not informative; rewrite this section.
Author Response
Dear Editor,
The coauthors and I greatly appreciated the reviewer's encouraging, critical, and constructive comments on this manuscript. The comments were highly detailed and beneficial in enhancing the manuscript.
Please see the attachment.

Round 2
Reviewer 2 Report
Authors addressed all comments suggested by me. Now this manuscript may be accepted in this esteemed journal.